# Secondary analysis of a James Lind Alliance priority setting partnership to facilitate knowledge translation in degenerative cervical myelopathy (DCM): insights from AO Spine RECODE-DCM

Benjamin Davies,[1] Jamie Brannigan [2] Oliver D Mowforth,[1] Danyal Khan,[3] Angus G K McNair [4,5] Lindsay Tetreault,[6,7] Iwan Sadler,[8] Ellen Sarewitz,[8] Bizhan Aarabi,[9] Brian Kwon,[10] Toto Gronlund,[11] Vafa Rahimi-Movaghar,[12] Carl Moritz Zipser [13] Peter John Hutchinson [14] Shekar Kurpad,[15] James S Harrop,[16] Jefferson R Wilson,[7] James D Guest,[17] Michael G Fehlings,[7] Mark R N Kotter[1]

For numbered affiliations see end of article.

**Correspondence to**
Dr Benjamin Davies;
bd375@cam.ac.uk

## ABSTRACT

**Objectives** To explore whether a James Lind Alliance Priority Setting Partnership could provide insights on knowledge translation within the field of degenerative cervical myelopathy (DCM).

**Design** Secondary analysis of a James Lind Alliance Priority Setting Partnership process for DCM.

**Participants and setting** DCM stake holders, including spinal surgeons, people with myelopathy and other healthcare professionals, were surveyed internationally. Research suggestions submitted by stakeholders but considered answered were identified. Sampling characteristics of respondents were compared with the overall cohort to identify subgroups underserved by current knowledge translation.

**Results** The survey was completed by 423 individuals from 68 different countries. A total of 22% of participants submitted research suggestions that were considered 'answered'. There was a significant difference between responses from different stakeholder groups (p<0.005). Spinal surgeons were the group which was most likely to submit an 'answered' research question. Respondents from South America were also most likely to submit 'answered' questions, when compared with other regions. However, there was no significant difference between responses from different stakeholder regions (p=0.4).

**Conclusions** Knowledge translation challenges exist within DCM. This practical approach to measuring knowledge translation may offer a more responsive assessment to guide interventions, complementing existing metrics.

## STRENGTHS AND LIMITATIONS OF THIS STUDY

⇒ A large number of stakeholders including patients, clinicians and researchers were surveyed, generating 76 research questions.
⇒ Responses came from individuals in 68 countries.
⇒ The protocol for collecting the data analysed in this study has been published previously.
⇒ The dissemination of the online survey through national organisations and research networks makes this study vulnerable to response bias.

dysfunction worldwide, affecting up to 2% of adults.[1 2] It arises when arthritic and/or congenital changes in the cervical spine cause progressive damage and injury to the cervical spinal cord. Today, despite treatment, most patients with DCM will be left with some disability. This is often due to missed or late diagnosis.[3] Furthermore, a recent comparative study demonstrated that people with DCM have among the lowest quality of life scores of chronic diseases.[4 5] Consequently, urgent progress is required.

To facilitate this, AO Spine Research objectives and Common Data Elements for DCM (RECODE-DCM) (aospine.org/recode), a multistakeholder consensus process was undertaken. This process aimed to accelerate research progress by defining key pieces of information which can better help individual studies deliver changes in care. It combined a number of consensus initiatives, including a James Lind Alliance (JLA) Priority Setting

## INTRODUCTION

Degenerative cervical myelopathy (DCM) is the most common cause of spinal cord

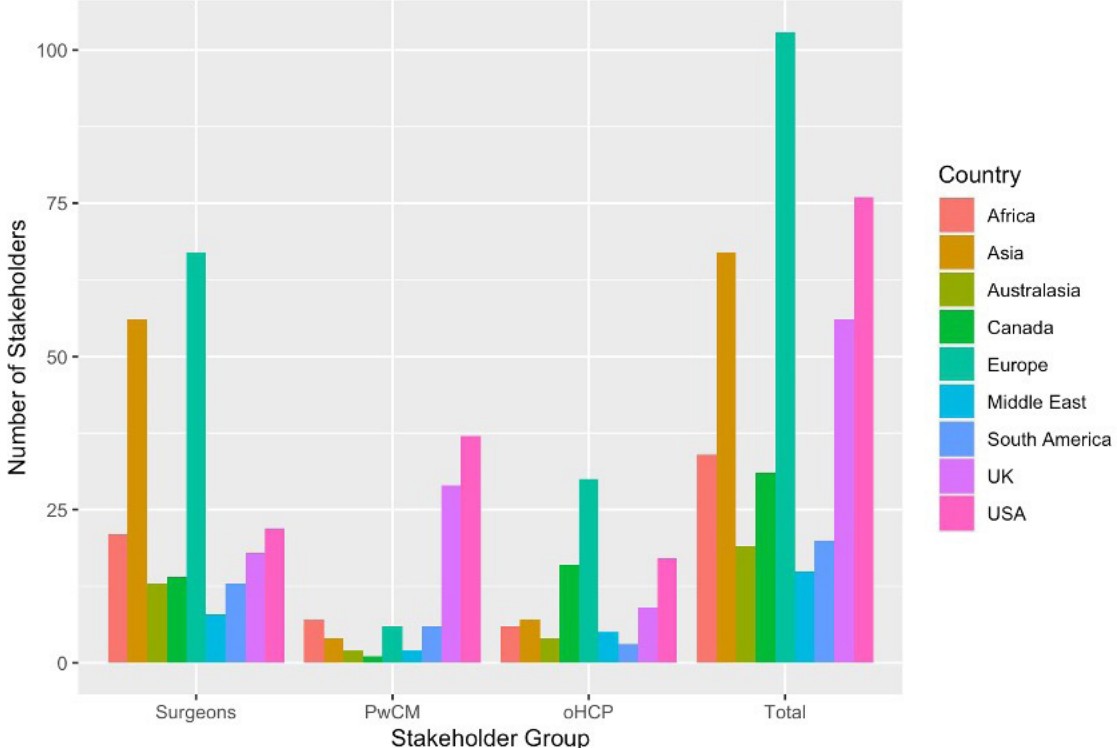

**Figure 1** Number of stakeholders that submitted questions, by region of stakeholder. Spinal surgeons were mostly based in Europe and Asia, while PwCM were much more likely to be from the UK or USA. pHCP, other healthcare professional; PwCM, persons with degenerative cervical myelopathy.

Partnership (PSP), to establish the top 10 research uncertainties.[6–8]

While an improvement in outcomes will require further scientific advance and clinical research, for individuals to benefit from any such progress, new knowledge must also transfer into clinical practice.[9] This transfer of knowledge, or knowledge translation (KT), is not straightforward,[10] and has been reported to take well over a decade in some cases.[11] For people with DCM, effective KT could be considered as important as knowledge discovery. This would be reflected in their selection of 'raising awareness' as the number one research priority for DCM.[12]

A variety of strategies and frameworks have been proposed to optimise the KT process,[13 14] including the formation of clinical practice guidelines. However, commonly, this process requires active surveillance and iteration. To that end, approaches to measure knowledge uptake are important but less well defined.

The aim of a PSP is to identify the critical knowledge gaps. This starts by seeking research suggestions from both people who have and who treat a condition (eg, DCM), across relevant healthcare disciplines. These submissions are then processed and consolidated into summary questions. Each summary question is evaluated against the current evidence base and is removed from the process if it is felt to have already been answered. The remaining questions are then taken forward to be prioritised.[8] These steps for AO Spine RECODE DCM have been previously described.[8]

Here, we explored the concept that the questions submitted by individuals as being 'ongoing research questions' but considered otherwise to have been answered, might highlight areas where KT was particularly lacking.

## METHOD
### Survey
The protocol for AO Spine RECODE-DCM is published in the Global Spine Journal.[15] DCM stakeholders were recruited to an internet survey hosted by Calibrum (California, USA). AO Spine RECODE-DCM identified three principal stakeholder groups to partake in this initiative: spinal surgeons, persons with DCM (PwCM) and their family or friends, and other healthcare professionals (oHCPs), including neurologists and physiotherapists.

A detailed summary of the dissemination process has been published.[7] An international contact directory was compiled of DCM stakeholder individuals and organisations. The directory comprised a list of names and contact email addresses for stakeholder individuals such as neurosurgeons, orthopaedic surgeons, neurologists, general practitioners and physiotherapists. Contact details for stakeholder organisations were also collected, including medical charities, universities, medical colleges, hospitals and medical journals. An email campaign targeted at stakeholders in the contact directory was executed using MailChimp (Georgia, USA). Emails provided a concise introduction to AO Spine RECODE-DCM, explained that we had identified the individual as someone who may be

**Table 1** Spinal surgeon stakeholders (N=232), subgroup analysis

|  | Unanswered (%) |  | Answered (%) |  | P value |
| --- | --- | --- | --- | --- | --- |
| Age | 44.4 |  | 44.5 |  | 0.6 |
| Male gender | 152 | 97 | 73 | 97 | 1 |
| Region |  |  |  |  | 0.21 |
| USA | 13 | 8 | 9 | 12 |  |
| UK | 11 | 7 | 7 | 9 |  |
| Canada | 12 | 8 | 2 | 3 |  |
| Europe | 49 | 31 | 18 | 24 |  |
| South America | 6 | 4 | 7 | 9 |  |
| Middle East | 3 | 2 | 5 | 7 |  |
| Asia | 40 | 25 | 16 | 21 |  |
| Australasia | 8 | 5 | 5 | 7 |  |
| Africa | 15 | 10 | 6 | 8 |  |
| Research cluster |  |  |  |  |  |
| DCM case treated yearly |  |  |  |  | 0.78 |
| 0–25 | 28 | 18 | 15 | 20 |  |
| 25–50 | 55 | 35 | 24 | 32 |  |
| 50–100 | 47 | 30 | 19 | 25 |  |
| 100+ | 27 | 17 | 16 | 21 |  |
| Year's experience | 13.6 | 9.7 | 14.2 | 13.5 |  |
| Neurosurgeon by training | 95 | 61 | 45 | 60 | 0.35 |
| From a high-activity, DCM research cluster | 43 | 27 | 16 | 21 | 0.41 |
| High-income country | 111 | 71 | 47 | 63 | 0.28 |

A high activity DCM research cluster was defined from a prior co-author network analysis—specifically DCM research activity clusters geographically to North America and East Asia (Japan, China and South Korea).
p<0.05
DCM, degenerative cervical myelopathy.

interested in participating, and provided a link to the survey. A total of five emails were sent to the global contact directory, each separated by approximately 1 week.

Respondents were randomised to a core outcome set stream and a PSP stream. In the PSP stream, participants were invited to enter as free text what they thought were the most important DCM research questions within each of the four categories of diagnosis, treatment, long-term care and follow-up and other.

The survey was closed at the point of information saturation, defined as no additional unique research suggestions at a 2-week interval. Following closure of the survey, research suggestions were processed by an information specialist.[8 15] Suggestions were grouped into common themes which were then used to form representative summary questions. All summary questions underwent an evidence checking process, including search of the literature and discussion with the Steering Committee. Questions were defined as either 'unanswered' or 'answered' depending on whether there was sufficient quality of evidence available in the literature. Scoping reviews of the literature were conducted by LT, the designated information specialist for this JLA PSP, to find evidence to support this process. Questions that were considered 'answered' were removed from the process following review and discussion with the steering committee, composed of 6 neurosurgeons, 1 orthopaedic surgeon, 2 neurologists, 1 primary care physician, 3 rehabilitation specialists and 12 PwCM.[8]

Of the 76 summary questions generated, 2 were considered to have been answered: (1) What is the safety and efficacy of surgical interventions for DCM? and (2) What is the efficacy and safety of anterior vs posterior surgery in patients with DCM? The decision to remove this latter question also considered that Cervical Spondylotic Myelopathy Surgical Trial (CSM-S, NCT02076113), a randomised controlled trial of anterior vs posterior surgery, was in process. For brevity these will be referred to as the 'effectiveness' and 'anterior versus posterior' questions. The remaining 74 questions, which were considered unanswered, are publicly available on the JLA PSP website.[16]

## Analysis

Demographics of participants who submitted 'answered' and 'unanswered' summary questions were aggregated for analysis. For HCPs, this included specialty, experience with DCM, age and country of employment. For PwCM

**Table 2** Comparison of respondent demographics of participants who submitted research suggestions that mapped to answered (N=95) compared with unanswered (N=328) summary questions

| | Unanswered (%) | | Answered (%) | | P value |
|---|---|---|---|---|---|
| N | 328 | 78 | 95 | 22 | |
| Stakeholder group | | | | | <0.005* |
| Spinal surgeons | 157 | 48 | 75 | 79 | |
| People with DCM and their supporters | 82 | 25 | 12 | 13 | |
| Other healthcare professionals | 89 | 27 | 8 | 8 | |
| Age (SD) | 47.9 | 11.7 | 46.4 | 11.68 | 0.25 |
| Male gender | 230 | 70 | 84 | 88 | <0.005 |
| Region | | | | | 0.4 |
| USA | 62 | 19 | 14 | 15 | |
| UK | 43 | 13 | 13 | 14 | |
| Canada | 27 | 8 | 4 | 4 | |
| Europe | 82 | 25 | 21 | 22 | |
| South America | 14 | 4 | 8 | 8 | |
| Middle East | 10 | 3 | 5 | 5 | |
| Asia | 49 | 15 | 18 | 19 | |
| Australasia | 13 | 4 | 6 | 6 | |
| Africa | 28 | 9 | 6 | 6 | |
| From a high-activity, DCM research cluster | 109 | 33 | 23 | 24 | 0.12 |
| High-income country | 250 | 76 | 63 | 66 | 0.07 |

A high-activity DCM research cluster was defined from a prior coauthor network analysis—specifically DCM research activity clusters geographically to North America and East Asia (Japan, China and South Korea).
*Significance, p<0.05.
DCM, degenerative cervical myelopathy.

or their supporters, this included country of residence and years lived with DCM. All participants were asked to provide their age and biological sex.

Geography is often an important consideration for KT for many reasons, including language, applicability to local practice and the physical barrier it can create for information exchange.[17] To explore this, country of residence or practice were aggregated into common zones—either by country if there was sufficient representation or continent if not. Countries were further categorised as higher-income countries or not, using the World Bank (worldbank.org) classification (22 October 2020). In addition, we and others have identified that DCM research is largely derived from two geographical clusters: North America (Canada and the USA) and East Asia (Japan, Korea and China).[17 18] To explore a relationship between research activity and KT, participants were also defined by whether they reside or practice within a research cluster or not.

Comparisons between groups, based on factors such as region and level of experience, were made using $\chi^2$ test for categorical or ordinal data, and Mann-Whitney U test for continuous data. Significance was defined as p<0.05.

Analysis and data visualisation were performed using R (V.4.0.5; R Core Team, 2020) and RStudio (V.1.4.1106; RStudio Team, 2021).

## Patient and public involvement

Patient and carer representatives were engaged throughout the process. They helped define the scope and were involved in the review of all patient-facing media. They were involved in all steering group meetings and decisions. They collaborated with patient organisations and helped to reach a diverse range of patient and carers groups for the surveys and final workshop. Patient representatives will help disseminate the PSP findings and work with patient and charitable organisations to develop discrete research questions from the final priorities to take forward for funding.

## RESULTS
### Summary
The survey was completed by 423 individuals from 68 different countries.[7] This included 232 surgeons (55%), 94 PwCM (22%) and 95 oHCP (23%). PwCM were principally from USA (41%) or the UK (32%). Surgeons and oHCP were more evenly distributed (figure 1).

In total, 95 (22%) participants submitted a research suggestion that mapped to one or both of these answered research questions; 51 (12%) 'effective' and 44 (10%) 'anterior versus posterior'. This included 75 (32%) spinal surgeons, 12 (13%) PwCM and 8 (8%) oHCPs.

**Table 3** Other healthcare professional stakeholders (N=95), subgroup analysis

|  | Unanswered (%) |  | Answered (%) |  | P value |
| --- | --- | --- | --- | --- | --- |
| Age | 45.7 |  | 42.8 |  |  |
| Male gender | 52 | 58 | 3 | 38 | 0.4 |
| Region |  |  |  |  |  |
| USA | 17 | 11 | 0 | 0 | 0.49 |
| UK | 7 | 4 | 2 | 3 |  |
| Canada | 14 | 9 | 2 | 3 |  |
| Europe | 28 | 18 | 2 | 3 |  |
| South America | 3 | 2 | 0 | 0 |  |
| Middle East | 5 | 3 | 0 | 0 |  |
| Asia | 6 | 4 | 1 | 1 |  |
| Australasia | 3 | 2 | 1 | 1 |  |
| Africa | 6 | 4 | 0 | 0 |  |
| Research cluster (Japan/China/North America) |  |  |  |  |  |
| DCM volume |  |  |  |  | 0.23 |
| 0–25 | 49 | 31 | 7 | 9 |  |
| 25–50 | 20 | 13 | 0 | 0 |  |
| 50–100 | 12 | 8 | 0 | 0 |  |
| 100+ | 7 | 4 | 1 | 1 |  |
| Years experience | 14.7 | 9.7 | 9.9 | 13.5 | 0.88 |
| From a high-activity, DCM research cluster | 32 | 36 | 2 | 25 | 0.8 |
| High-income country | 72 | 81 | 6 | 75 | 1 |
| Discipline |  |  |  |  |  |
| Neurologist | 18 | 11 | 0 | 0 | 0.23 |
| Physiotherapist | 10 | 6 | 1 | 1 |  |
| Rehabilitation medicine | 9 | 6 | 3 | 4 |  |
| General practitioner | 9 | 6 | 0 | 0 |  |
| General physician | 10 | 6 | 1 | 1 |  |
| Other | 32 | 20 | 3 | 4 |  |

A high-activity DCM research cluster was defined from a prior coauthor network analysis—specifically DCM research activity clusters geographically to North America and East Asia (Japan, China and South Korea).
p<0.05
DCM, degenerative cervical myelopathy.

## Submission of research suggestions that were 'answered' versus 'unanswered'

In the group that submitted a research suggestion that was deemed to be 'answered' (ie, around surgical 'effectiveness' and 'anterior vs posterior' surgery), there were 75 (79%) surgeons, 12 (13%) oHCP and 8 (8%) PwCM. Spinal surgeons (p<0.005) and those of male sex (p<0.005) were more likely to submit a research suggestion that was already answered (table 1; online supplemental material 1).

Individuals were less likely to submit an answered research question if they resided or practised within an active DCM research cluster (Japan, China, South Korea, USA or Canada) or high-income countries (tables 1–4).

Professional experience or discipline was not associated with the likelihood of submitting an answered research question. Of note, no neurologist (N=18) submitted a research suggestion that mapped to an 'answered' research question (table 3).

Demographics were compared of those who submitted answered research suggestions, by whether it mapped to the 'effectiveness' or to the 'anterior versus posterior' questions. Spinal surgeons and respondents from Asia (p<0.05) were more likely to submit research questions related to 'anterior versus posterior' approaches.

## DISCUSSION

KT is a major issue in DCM. This is reflected by its selection as the number one research priority by AO Spine RECODE DCM-Raising Awareness.[12] This was also reflected within this analysis, as 22% of participants

**Table 4** Persons with DCM or their supporters (friends or family), subgroup analysis

| | Unanswered (%) | | Answered (%) | | P value |
|---|---|---|---|---|---|
| Age | 45.7 | | 42.75 | | 0.47 |
| Male gender | 52 | 63 | 3 | 38 | 0.03 |
| Region | | | | | |
| USA | 32 | 39 | 5 | 43 | 0.96 |
| UK | 25 | 30 | 4 | 33 | |
| Canada | 1 | 1 | 0 | 1 | |
| Europe | 5 | 6 | 1 | 7 | |
| South America | 5 | 6 | 1 | 7 | |
| Middle East | 2 | 2 | 0 | 3 | |
| Asia | 3 | 4 | 1 | 4 | |
| Australasia | 2 | 2 | 0 | 3 | |
| Africa | 7 | 9 | 0 | 9 | |
| Years lived with DCM | 5.5 | 4.8 | 5.3 | 4.2 | 0.8 |
| From a high-activity, DCM research cluster | 34 | 41 | 10 | 45 | 1 |
| High-income country | 67 | 82 | 10 | 89 | 1 |

A high-activity DCM research cluster was defined from a prior coauthor network analysis—specifically DCM research activity clusters geographically to North America and East Asia (Japan, China and South Korea).
p<0.05
DCM, degenerative cervical myelopathy.

submitted research suggestions that were considered 'answered'. Spinal surgeons were more likely to submit an answered research question than oHCPs or PwCM. Anterior versus posterior surgery was more likely to be suggested by surgeons and respondents from Asia. Individuals living or practising within a higher-income country, or a country with high DCM research activity, tended to be less likely to submit an answered research suggestion; this association, however, was non-significant.

### Can evidence checking of research suggestions act as a KT metric and inform KT strategy?

Ultimately, this was an exploratory analysis of an existing dataset, and cannot establish whether analysis of research suggestions is truly an effective KT metric. For example, many respondents in possession of the evidence may have considered the 'anterior versus posterior' question to be unanswered. Our findings may instead reflect conflicting interpretations of the evidence, rather than poor KT. In this regard, it was perhaps noteworthy that this question was more likely to be submitted by Asian surgeons where OPLL is more prevalent.[19] However, the results, taken in wider context, suggest promise.

Building on the significant growth in DCM research[20] and clinical evidence,[21 22] clinical practice guidelines for DCM have been developed by AO Spine[23] and the World Federation of Neurosurgeons separately.[24–27] While there remain many unanswered questions in DCM,[15] these guidelines consolidate the current evidence on the effectiveness of surgical treatment.[23] Guidelines are considered one of the most effective tools for KT.[28 29] Despite this, an audit of surgical practice has shown poor adherence

to these guidelines,[30] and DCM research continues to be dominated by investigations into these 'answered' research questions by surgeons.[31] This would suggest an ongoing KT gap and would align with the observed 75 (33%) surgeons who submitted at least one 'answered' question relating to this. This would also align with the on average 10–15 years[11] taken to bring new knowledge into routine practice.

Efforts to support the dissemination of these guidelines and inform evidence-based care are ongoing. One of the challenges is the large number of specialities currently coordinating DCM care—all potential target audiences, for example, general practice, neurology, physiotherapy, orthopaedics, neurosurgery, rheumatology, gerontologists and rehabilitation physicians.[32] Although the scope of answered questions was restricted to surgery, these research suggestions were still submitted by 8% of oHCP and 13% of PwCM; 8% and 13%.

The success of KT, or strategies to accelerate KT, are conventionally assessed through changes to guidelines, surveys of care providers and measurement of service/product/pathway adoption, where applicable.[33] While valid, each of these metrics takes considerable time to perform and, often, for example, with respect to uptake within guidelines, would lag considerably a KT intervention. This means the recognition of successful or failed strategies, and/or need to iterate KT strategies, often is not very responsive.

Our experience here would suggest the assessment of the 'answered' research suggestions could offer a live snapshot of KT progress, concerning both patients

and clinicians. Clearly its application may not be appropriate in all settings. For example, this analysis approach could not be applied to a PSP for perioperative care in Canada, where no research suggestions were deemed to be answered already.[34] This approach will also be vulnerable to selection bias. For example, as was the case in AO Spine RECODE DCM, engaging stakeholders through electronic surveys outside of spinal surgery was very difficult. However, efforts could be taken to mitigate this.

Further, the relative differences in sampling characteristics may be helpful. In this study, participants were less likely to submit an answered research question if they came from a high-income country, or a country with higher DCM research activity. Questions relating to anterior versus posterior surgery were also more likely to come from Asia. Overall, these differences may indicate groups underserved by current KT strategies.

### Factors contributing to KT gaps in DCM

There are several proposed barriers to rapid dissemination of DCM knowledge:

Terminology—First proposed in 2015,[35] the umbrella term of DCM is still not universal.[36] 'CSM' is the most commonly used term, but this has an inconsistent definition.[36] The use of variable terminology may therefore impede KT.

Geography/language—The major international guidelines have been published in English.[23] While this may not affect our study this affects international adoption of new knowledge.[37]

Adaptation for local use—Adaptation of knowledge to a local context is a key step in the knowledge-to-action cycle.[38] This is a proactive process which must take place in individual hospitals and hospital networks.[39]

There are also further barriers to the transfer of knowledge between different stakeholder groups.

For oHCPs, 'knowledge silos' have been described over the last decade between specialties.[40 41] The existence of different journals, vocabulary, professional organisations and priorities are all believed to contribute to this.[40 41] Silos act to from closed-communication loops which inhibit knowledge diffusion. We expect that this applies to surgeons, neurologists and general practitioners in the case of DCM. Impaired collaboration also exists between clinicians and allied health professionals,[42] which may form a barrier to KT within specialties.

KT to PwCM is also significant. If considering knowledge about surgical interventions, it has been long established that improved patient knowledge in the preoperative phase enhances postoperative outcomes.[43] This is true in several domains, including postoperative compliance[44] and subjective pain reporting.[45] Barriers to the transfer of knowledge to PwCM include clinician knowledge and health literacy.[46]

The relative comparison between 'answered' questions submitted by different stakeholder groups may reveal further insights about KT and the strategies used to tackle the above barriers. It is hoped the emergence of the RECODE-DCM community may also become a tool to address this.[47]

### Conclusion and future directions

Answered research questions were frequently submitted during the AO Spine RECODE DCM PSP, indicating a KT problem in DCM. This practical approach to measuring KT may more widely offer a responsive assessment to guide interventions, complementing existing KT metrics which provide retrospective assessments. In the future, KT in AOSpine RECODE-DCM will need to involve considerable outreach to the broader community of healthcare providers involved with DCM, healthcare funders and policy-makers and the public.

**Author affiliations**
[1]Department of Clinical Neurosurgery, University of Cambridge, Cambridge, UK
[2]School of Clinical Medicine, University of Cambridge, Cambridge, UK
[3]Queen Square Institute of Neurology, University College London, London, UK
[4]Centre for Surgical Research, Bristol Medical School, Population Health Sciences, University of Bristol, Bristol, UK
[5]Department of General Surgery, North Bristol NHS Trust, Bristol, UK
[6]Department of Medicine, University College Cork, Cork, Ireland
[7]Division of Neurosurgery and Spine Program, Department of Surgery, University of Toronto, Toronto, Ontario, Canada
[8]Myelopathy.org, Cambridge, UK
[9]Division of Neurosurgery, University of Maryland, Baltimore, Maryland, USA
[10]Division of Spine Surgery, Vancouver General Hospital, University of British Columbia, Vancouver, British Columbia, Canada
[11]National Institute for Health Research, University of Southampton, Southampton, UK
[12]Academic Department of Neurological Surgery, Sina Trauma and Surgery Research Center, Tehran, Iran (the Islamic Republic of)
[13]Department of Neurology, University Hospital Balgrist, Zurich, Switzerland
[14]Department of Academic Neurosurgery, University of Cambridge, Cambridge, UK
[15]Division of Neurosurgery, Medical College of Wisconsin, Milwaukee, Wisconsin, USA
[16]Division of Neurosurgery, Thomas Jefferson University Hospital, Jefferson Health System, St Louis, Mississippi, USA
[17]Department of Neurosurgery and The Miami Project to Cure Paralysis, The Miller School of Medicine, University of Miami, Miami, Florida, USA

**Acknowledgements** The research priorities were organised and funded by AO Spine through the AO Spine Knowledge Forum Spinal Cord Injury, a focused group of international Spinal Cord Injury experts. AO Spine is a clinical division of the AO Foundation, which is an independent medically guided not-for-profit organisation. Study support was provided directly through the AO Spine Research Department. This research aligns with the AO Spine RECODE-DCM James Lind Alliance top research priority Raising Awareness selected by people living and working with DCM. For further information on how this process was conducted, why this question was prioritised, and global updates on currently aligned research, please visit aospine. org/recode/raising-awareness. We thank Dr Joanna Crocker (Interdisciplinary Research in Health Sciences, University of Oxford) for her help with our methodology and statistical reporting. AM is funded by a Clinician Scientist Fellowship (NIHR-CS-2017-17-010) from the UK National Institute for Health Research (NIHR) and supported by the NIHR Biomedical Research Centre at the University Hospitals Bristol NHS Foundation Trust and the University of Bristol. MRNK is supported by a NIHR Clinician Scientist Award, CS-2015-15-023. BMD is supported by a research fellowship from the Royal College of Surgeons (London), and the NIHR, UK.

**Contributors** BD, ODM, DK, AGKM, LT, IS, ES, BA, BK, TG, VR-M, CMZ, PJH, SK, JSH, JRW, JDG, MGF and MRNK were involved in the conceptualisation and design of this study. BD and JB wrote the first draft of the manuscript. BD, JB, ODM, LT, BK and MGF were involved in revising the manuscript. BD, JB, ODM, DK, AGKM, LT, IS, ES, BA, BK, TG, VR-M, CMZ, PJH, SK, JSH, JRW, JDG, MGF and MRNK reviewed and approved the final manuscript.

**Funding** NIHR Clinician Scientist Fellowship, Award/Grant no: NIHR-CS-2017-17-010 (AM). NIHR Clinician Scientist Award, Award/Grant no: CS-2015-15-023 (MRNK). NIHR Research Fellowship, Award/Grant no: NA (BMD).

**Disclaimer** The views expressed in this publication are those of the authors and not necessarily those of the NHS, the National Institute for Health Research or the Department of Health.BD is the guarantor for the overall content of this work, accepting full responsibility for the study. BD had access to the data, and controlled the decision to publish.

**Competing interests** BD, JB, ODM, IS, ES, MGF and MRNK have voluntary roles at Myelopathy.org, an international DCM charity.

**Patient and public involvement** Patients and/or the public were involved in the design, or conduct, or reporting, or dissemination plans of this research. Refer to the Methods section for further details.

**Patient consent for publication** Not applicable.

**Ethics approval** Ethical approval for the study was granted by the Human Biology Research Ethics Committee, University of Cambridge (HBREC.2019.14). All patients involved provided informed consent.

**Provenance and peer review** Not commissioned; externally peer reviewed.

**Data availability statement** Data are available on reasonable request.

**ORCID iDs**
Jamie Brannigan http://orcid.org/0000-0003-1697-403X
Angus G K McNair http://orcid.org/0000-0002-2601-9258
Carl Moritz Zipser http://orcid.org/0000-0002-4396-4796
Peter John Hutchinson http://orcid.org/0000-0002-2796-1835

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
