## [Reviewer comments · BMJ Open]

ARTICLE DETAILS

TITLE (PROVISIONAL)	Secondary analysis of a James Lind Alliance priority setting partnership to facilitate knowledge translation in degenerative cervical myelopathy (DCM): Insights from AO Spine RECODE-DCM
AUTHORS	Davies, Benjamin; Brannigan , Jamie; Mowforth, Oliver; Khan, Danyal; McNair, Angus; Tetreault, Lindsay; Sadler, Iwan; Sarewitz, Ellen; Aarabi, Bizhan; Kwon, Brian; Gronlund, Toto; Rahimi-Movaghar, Vafa; Zipser, Carl Moritz; Hutchinson, Peter; Kurpad, Shekar; Harrop, James; Wilson, Jefferson R.; Guest, James; Fehlings, Michael G.; Kotter, Mark R.N.

VERSION 1 – REVIEW

REVIEWER	Marie-Hardy, Laura Hopital Universitaire Pitie Salpetriere, Orthopaedic Surgery
REVIEW RETURNED	27-Aug-2022

GENERAL COMMENTS	This is an interesting study with rigourous method to assess medical and patients knowlegde regarding DCM, indeed a matter of public health. The method is rigourous and the study weel-written. I would have add further explanations on how is a question considered "answered" and add as addiational documents the the 76 summary questions generated. I am in favor of accepting this study for publication after these minor revisions.
---

REVIEWER	Koyanagi, I Hokkaido Neurosurgical Memorial Hospital, Sapporo, Japan
REVIEW RETURNED	13-Nov-2022

GENERAL COMMENTS	The authors reported the results of the survey by 423 individuals from 68 countries regarding DCM research questions. They analyzed distributions and factors of responders submitting "answered" suggestions, effectiveness of surgery and anterior vs posterior approaches. This is an interesting analysis based on the well-organized international survey, and will help to understand current research issues of DCM. Followings can be pointed out for some revisions. 1) This manuscript contains two sets of Table 1 and Table 2. Probably the second Table 1 will be Table 3, and the second Table 2 will be Table 4. The second Table 2 may be incomplete, such as row of Age showing only in "unanswered" column. 2) This survey generated 76 summary questions and two were "answered". Some brief descriptions on 74 "unanswered" questions may help readers to understand this study.
---

REVIEWER	Sharma, Ayush
-----------------	---------------

	Central Railway Hospital Byculla
REVIEW RETURNED	20-Nov-2022

GENERAL COMMENTS	Although its a very interesting read but it doesn't have a significant impact on the present knowlage or to the future direction of research on DCM.
--

VERSION 1 – AUTHOR RESPONSE

Point for Point Response

R = reviewer comment

A = author response

R: This is an interesting study with rigorous method to assess medical and patients knowledge regarding DCM, indeed a matter of public health. The method is rigorous and the study well-written. I would have added further explanations on how is a question considered "answered" and add as additional documents the 76 summary questions generated. I am in favour of accepting this study for publication after these minor revisions.

A: We thank the reviewer for reviewing our manuscript and for their kind words. We have expanded our methods section to explain in more detail how a question was considered 'answered'. We have also added a citation which will bring the user to the James Lind Alliance webpage where all 76 summary questions are listed in a PDF.

It is our hope that this study is a springboard into further work to improve undergraduate neurosurgical education.

R: The authors reported the results of the survey by 423 individuals from 68 countries regarding DCM research questions. They analyzed distributions and factors of responders submitting "answered" suggestions, effectiveness of surgery and anterior vs posterior approaches. This is an interesting analysis based on the well-organized international survey, and will help to understand current research issues of DCM. Followings can be pointed out for some revisions.

1) This manuscript contains two sets of Table 1 and Table 2. Probably the second Table 1 will be Table 3, and the second Table 2 will be Table 4. The second Table 2 may be incomplete, such as row of Age showing only in "unanswered" column.

2) This survey generated 76 summary questions and two were "answered". Some brief descriptions on 74 "unanswered" questions may help readers to understand this study.

A: We thank the reviewer for reviewing our manuscript and for the helpful suggestions for improvement. In response to the first point, we have now numbered our figures appropriately. We thank the reviewer for noticing an omission of mean age data in table 2, which we have now added. In response to the second point, it was regrettably not possible to provide brief descriptions of the 74 unanswered questions due to the variety of questions proposed. Instead, we have added a citation which will bring the user to the James Lind Alliance webpage where all 76 summary questions are listed in a PDF. We hope that this is a suitable solution.

R: Although it's a very interesting read but it doesn't have a significant impact on the present knowledge or to the future direction of research on DCM.

A: We thank the reviewer for reviewing our manuscript and for their feedback. We regret that the reviewer did not value the significance of our findings. We believe that the identification of knowledge translation gaps and their determinants not only presents an opportunity to improve translation in DCM, but more generally this approach can complement knowledge translation metrics when applied to other diseases.

VERSION 2 – REVIEW

REVIEWER	Koyanagi, I Hokkaido Neurosurgical Memorial Hospital, Sapporo, Japan
REVIEW RETURNED	25-Dec-2022
GENERAL COMMENTS	The authors revised almost correctly according to reviewer's comments. Please correct wrong number of tables: Table 1 of page 14 will be Table 3, and Table 2 of page 16 will be Table 4.

VERSION 2 – AUTHOR RESPONSE

Point for Point Response

R = reviewer comment

A = author response

R: Please include any relevant statistical results in the results section of the Abstract.

A: We thank the reviewer for reviewing our manuscript and for their feedback. We have expanded our abstract to include any relevant statistical results.

R: Although we appreciate that the protocol has been previously published please provide more detail on the Methods used, e.g. how were participants recruited?

A: We thank the reviewer for reviewing our manuscript and for the helpful suggestion for improvement. We have included more detailed information regarding our methods used, paying particular attention to participant recruitment.

R: Please clarify whether participants gave consent to participate in your ethics statement in the main text of the manuscript.

A: We thank the reviewer for reviewing our manuscript and highlighting this important point. We have provided further elaboration of our ethics statement to state that all participants provided informed consent.